# Endoscopic Endonasal Repair of Congenital Choanal Atresia: Predictive Factors of Surgical Stability and Healing Outcomes

**DOI:** 10.3390/ijerph19159084

**Published:** 2022-07-26

**Authors:** Salvatore Ferlito, Antonino Maniaci, Alberto Giulio Dragonetti, Salvatore Cocuzza, Jerome Rene Lechien, Christian Calvo-Henríquez, Juan Maza-Solano, Luca Giovanni Locatello, Sebastiano Caruso, Francesco Nocera, Andrea Achena, Niccolò Mevio, Gabriella Mantini, Giorgio Ormellese, Angelo Placentino, Ignazio La Mantia

**Affiliations:** 1Department of Medical and Surgical Sciences and Advanced Technologies “GF Ingrassia”, E.N.T. Section, University of Catania, 95124 Catania, Italy; ferlito@unict.it (S.F.); s.cocuzza@unict.it (S.C.); sebastiano.caruso2404@gmail.com (S.C.); ciccionocera94@gmail.com (F.N.); igolama@gmail.com (I.L.M.); 2Rhinology Study Group of the Young-Otolaryngologists of the International Federations of Oto-rhino-laryngological Societies (YO-IFOS), 75009 Paris, France; jerome.lechien@umons.ac.be (J.R.L.); christian.calvo.henriquez@gmail.com (C.C.-H.); juan.maza.solano@gmail.com (J.M.-S.); locatello.lucagiovanni@gmail.com (L.G.L.); nicmevio@yahoo.it (N.M.); 3Otolaryngology Unit, 9338 Ospedale Niguarda “Ca’ Granda”, 20162 Milano, Italy; dottdragonetti@gmail.com (A.G.D.); andrea.achena@ospedaleniguarda.it (A.A.); gabriella.mantini@ospedaleniguarda.it (G.M.); giorgio.ormellese@ospedaleniguarda.it (G.O.); angelo.placentino@ospedaleniguarda.it (A.P.); 4Service of Otolaryngology, Foch Hospital, University of Paris Saclay, 91190 Paris, France; 5Service of Otolaryngology, Hospital Complex of Santiago de Compostela, 15701 Santiago de Compostela, Spain; 6Rhinology and Skull Base Unit, Department of Otorhinolaryngology, University Hospital Virgen de la Macarena, 41009 Seville, Spain; 7Department of Otorhinolaryngology, Sant’Antonio Abate Hospital, Azienda Sanitaria Universitaria Friuli Centrale, 33100 Tolmezzo, Italy

**Keywords:** choanal atresia, transnasal approach, surgical predictors, endoscopic repair, long-term outcomes

## Abstract

Background: To assess the long-term outcomes and independent predictors of surgical success of a one-stage minimally invasive surgical procedure for congenital choanal atresia (C.C.A.). Methods: a retrospective multicentric study was conducted between 2010 and 2022. An endonasal endoscopic approach was performed in 38 unilateral or bilateral C.C.A. children. All the patients were clinically and radiologically assessed and followed for at least 2 years. Seven outcome measures were applied. Consequently, surgical success was correlated with all the independent variables reported. Results: 18/38 (47.36%) patients presented normal postoperative healing, 8/38 (21.05) had moderate restenosis (<50%), while 12/38 (31.57%) cases were severe (>50%), requiring a surgical revision. No statistical significance was found for average hospital stay between stenosis >50% and <50% patients (*p* = 0.802) and postoperative pain (*p* = 0.075); instead, the severe restenosis group demonstrated a higher delay of breast suction (*p* < 0.001). Among the independent variables predictors of surgical success, the presence of Charge syndrome and rhinopharyngeal stenosis demonstrated higher risks for surgical revision (OR: 4.00, 95% CI: 0.57–28.01, and OR: 2.75, 95% CI: 0.55–13.69, respectively). On the contrary, the hypoplastic inferior turbinate and bilateral C.C.A. showed a lower risk for severe restenosis by a higher endoscopic surgical space and creating a single larger opening (OR: 0.88, 95% CI: 0.22–3.52, and OR: 0.45, 95% CI: 0.10–2.08). Conclusion: Several independent variables could influence the surgical success after C.C.A. endoscopic repair; however, more high-quality evidence is needed to generate an effective predictive model.

## 1. Introduction

Congenital choanal atresia (C.C.A.) is a rare craniofacial abnormality (1:5000/1:8000 new-borns), characterized in 71% of subjects by mixed bony and membranous obstructions while in 29% of cases, only bone is found [1,2]. This malformation could occur uni-laterally in 60% of cases or bilaterally in 40% of cases, and it is attributed to neural crest cell migration and mesodermal disorders [3]. Frequently, C.C.A. is associated with other congenital anomalies, such as CHARGE, Crouzon, Di George, Treacher-Collins, or Down syndrome [4]. Although bilateral C.C.A. may present with acute respiratory distress and require early intubation at birth, unilateral C.C.A. is characterized by persistent nasal discharge with chronic, foul-smelling, and unilateral rhinorrhea, with a consequently delayed diagnosis in adulthood [5,6,7,8]. The primary objectives of surgical treatment are to create as soon as possible a stable and adequate neochoanal orifice for regular breathing, to re-establish regular feeding for the child, and to reduce the postoperative hospital stay [9,10,11]. Although various surgical approaches have been suggested for treating C.C.A., such as the transnasal, the sublabial-transnasal, the transpalatine, or the transseptal approaches, nowadays, most C.C.A.s are operated on endoscopically. Indeed, minimally invasive approaches through transnasal endoscopic surgery allow access to the posterior nasal portion avoiding the limitations and complications of all previous approaches.

Nevertheless, often the technique requires revision surgery, mainly due to restenosis from scar or granulation tissue formation. In this regard, nasal stenting or the topical application of mitomycin is still controversial in the literature [12,13,14,15]. Strychowsky et al. in 2015 show that similar success rates for bilateral C.C.A. repair were reported with and without nasal stents [15]. The use of nasal stenting should probably be considered in the context of revision surgery or bilateral stenosis in syndromic patients while, in isolated congenital C.C.A., the stentless technique might be preferred [12]. In order to add evidence in the field, we retrospectively assessed the perceived postoperative functional and subjective outcomes in a series of C.C.A. patients, comparing the results obtained in the unilateral forms with the bilateral ones.

## 2. Materials and Methods

A retrospective multicentric study was performed at our ear, nose, and throat (E.N.T.) Units from 2010 to 2022. We followed the Strengthening the Reporting of Observational Studies in Epidemiology [16]. We initially considered for study including all the patients with congenital choanal atresia (C.C.A.) referred to our E.N.T. Units for endoscopic surgical repair. Data collection was performed by reviewing medical records. The study design is summarized in Figure 1.

All the subjects were children reporting breathing or breast-suction disorders. Patients with unilateral choanal atresia were subjected to clinical assessment for chronic foul-smelling and unilateral rhinorrhea. On the contrary, subjects with bilateral C.C.A. presented respiratory distress at birth, intermittent cyanosis, and difficulties relieved once crying. These patients were preventively subjected to oral feeding through an orogastric tube.

After failure to pass a catheter through the nasal cavity into the oropharynx, the diagnosis of choanal atresia was thus confirmed via clinical and endoscopic nasal assessment. Preoperative high-resolution spiral C.T. scans were performed through the bone algorithm, orienting perpendicularly to the hard palate plane. A C.T. section parallel to the posterior hard palate at the level of the pterygoid plates was obtained in all children. The sections were 0.5 to 1 mm thick. The width of the vomer at the posterior end and the width of the airspace of the choanae or membranous atresia were measured based on the measurement of the dimension from the lateral wall of the nasal cavity to the edge of the vomer of the posterior end.

Exclusion criteria from the study were partial or incomplete data recorded for C.C.A. and subjects lost to follow-up assessment. Dependent variables such as age at diagnosis, gender, atretic plate type, side of atresia, unilateral vs. bilateral, locoregional associated disorders such as turbinate hypoplasia, rhynopharyngeal web and stenosis, surgical details, postoperative outcomes, surgical complications, recurrence rate as severe restenosis (>50%), and need for revision surgery were evaluated [3,17].

### 2.1. Surgical Technique and Postoperative Management

All the endoscopic surgical procedures were performed under general anesthesia, using 0° endoscopes (2.7 mm Karl Storz, Germany) connected to a high-definition system (Karl Storz, Germany). We performed a mucosal decongestion using cotton pledgets soaked through a saline solution with xylometazoline and lidocaine (5 mL and 2 mL/10 mL). Two different surgeons performed the procedures.

Since bilateral forms lead to neonatal respiratory distress, management has included the use of an oropharingeal (Guedel type) or orotracheal intubation. In all cases of single- or bilateral C.C.A., we used the mucoperiosteal flaps technique, consisting of an incision and elevation of the flap at the nasal mucosa conjunction between the vomer and the atretic plate. The posterior nasopharynx was protected during the maneuver with cotton inserted through the contralateral choana or oropharynx of the patient to avoid possible subperiosteal detachment of the posterior cul-de-sac during perforation of the atresal wall. The atretic plate portion was perforated with a skeeter at the inferior-posterior portion. Thus, the enlargement of the inferomedial atretic portion was performed with the microdebrider for the fibrous component (2.9 mm, Medtronic Inc., Jacksonville, FL, USA), while for the bone frame with a bite forceps and a drill (4.0 mm, Medtronic Inc., Jacksonville, FL, USA). Enlargement of the bony structure was continued with the drill up to the roof of the nasopharynx and the sphenoidal rostrum at the top while laterally bringing the pterygopalatine block at the level of the lateral nasopharyngeal wall; however, careful attention was paid to the area of the sphenopalatine foramen to avoid injuring the sphenopalatine artery. If necessary, we introduced the drill into the contralateral nasal cavity to improve surgical visual control. Consequently, we carried out the procedure using retrograde forceps or eventually with the drill or the microdebrider. Subsequently, the mucoperiosteal flap was used to cover the lateral wall of the nasal cavity, the pterygoid plates, and the medial side of the neochoana and septum.

In the presence of rhinopharyngeal stenosis or web under endoscopic visualization, a flap knife was used to cut the scarred mucosa on the posterior septum, on the posterior end of the inferior or middle turbinate, then excised with a shaver and Blakesley’s forceps.

### 2.2. Nasal Stenting

Stenting was routinely performed only after surgical correction of atresia for bilateral C.C.A. or revision surgery due to restenosis.

In contrast, isolated unilateral C.A.A.s did not undergo routine stenting. In unilateral patients, we placed a silicone fashioned stent (Foley Catheters, Medtronic Inc., Jacksonville, FL, USA) (4 to 6 weeks) on the affected side and attached it to the columella with a suture. Bilateral C.C.A. patients, on the other hand, were treated with a modified endotracheal tube (4 to 6 weeks) with four holes in the back (Portex polyvinyl chloride 3.0–3.5, Portex Ltd., Kent, UK) of the neochoana, allowing both breathing and saline irrigation at follow-up.

Nasal stenting was maintained until it failed to remain pervious under surrounding pressure, reducing the risk of restenosis or when an increased risk of infection, granulations formation, excessive nasal crusts, and persistent nasal discharge was detected. Moreover, the time was shorter in unilateral choanal atresia, fixed to the columella, and longer in bony plates, bilateral type, or revision surgeries.

### 2.3. Follow-Up and Postoperative Outcomes

Postoperative pediatric management included continuous positive airway pressure (CPAP) up to 48 h. Oral feeding was administered in all patients after complete wakefulness, while an intraoral orogastric feeding tube was placed in case of overt dysphagia. Each patient was assessed according to age-related pain scales and subjected to subsequent pain therapy with acetaminophen according to guidelines [18]. Proton-pump inhibitors were prescribed for 1 month. All patients received postoperative antibiotic prophylaxis for 7–10 days after surgery, regardless of stent use. All the patients performed nasal rinses postoperative for 1 month. We performed endonasal in-office medications 7, 14, and 30 days after surgery. Consequently, endoscopic evaluations were indicated at 45 and 90 days and yearly thereafter. Eustachian tube function was examined by tympanometry at each control visit [19]. The postoperative outcomes assessed for the healing process were the rate of restenosis, synechiae, granulation tissue formation, and the rate of reintervention procedures required after primary surgery. We considered neochoanal patency if lumen ≥ 50% without nasal symptoms. Thus, we subdivided all the patients into 4 classes: 0 = normal, no lumen restriction; 1 = limited restriction, lumen < 25%; −2 = partial restenosis, lumen between 25% and 50%; −3 = severe restenosis, lumen > 50%.

### 2.4. Statistical Analysis

Standard descriptive statistics were used, reporting mean and standard deviation for continuous variables and percentages for categorical ones. The independent *t*-test was performed for the normally distributed values, while the Mann–Whitney U test was performed for the non-normally distributed values. The chi-square test was performed to test the observed and expected data difference. The outcome odds were calculated and modeled as a combination of the predictor variables. Results were reported as odds ratios (OR) with 95% confidence intervals (CI).

Consequently, a Forest Plot diagram with all independent variables was generated. Due to the small sample size, the Jaccard coefficient between the dichotomous variables was calculated to observe the similarity between variables. The Jaccard coefficient J measures the similarity between sets of finite samples on binary (and non-binary) variables. It is defined as the size of the intersection divided by the size of the union of the sample sets:J(A,B) = |A∩B|/|A∪B|   0 ≤ J(A,B) ≤ 1

It is observed that values of J < 0.5 between outcomes and variables correspond to the protective factors (OR < 1), and values greater than 0.5 (greater similarity) represent the risk factors (OR > 1). The Kruskal–Wallis H test was used to assess subgroup differences in non-parametric features. A value of *p* < 0.05 was deemed to be statistically significant. All analyses were performed using the Social Sciences Statistical Program (IBM SPSS Statistics for Windows, I.B.M. Corp. Released 2017, Version 25.0 Armonk, NY, USA: I.B.M. Corp). The violin plots were performed using GraphPad Prism software (GraphPad Software Inc., Version 9, San Diego, CA, USA). Violin plots represented median as black bars, quartiles with upper and lower borders in the white box, while the density estimation was expressed as more concentrated data with the fatter the image.

## 3. Results

A total of 38 patients were included in the analysis, of which 24/38 (63.15%) isolated unilateral C.C.A. while 14/38 (36.84%) had bilateral C.C.A. (Table 1). The mean follow-up time was 2.93 ± 0.45 years. Associated upper airways congenital disabilities were observed in 33/38 (86.8%). In detail, Hypoplastic inferior turbinate was found in 23/38 (60.5%) cases, nasopharyngeal stenosis in 8/38 (21.1%), and rhinopharyngeal web in 2/38 (5.3%). Instead, we found 5/14 (%) syndromic patients, all CHARGE syndrome with bilateral C.C.A. The mean length of postoperative stenting was 38.4 ± 9.56 days, 44.3 ± 6 for bilateral C.C.A. patients, and 33.5 ± 4.94 for unilateral C.C.A. revision cases (*p* = 0.134).

Postoperative healing was overall normal at follow-up in 18/38 (68%) patients, including 7/38 (24%) bilateral and 11/37 (44%) unilateral C.C.A. cases (*p* = 0.804). A nonstatistical difference was also recorded for limited scar formation–partial restenosis (<50%), found in 8/38 (20%) subjects, including 4 (12%) bilateral and 4 (8%) unilateral (*p* = 0.385). In addition, severe restenosis (>50%) was reported in 12 (8%) cases, including 3 (4%) bilateral and 9 (4%) unilateral (*p* = 0.303), finally requiring surgical revision.

All the sinonasal-associated defects are summarized in Table 1.

The average delay in starting breast suction was 1.2 ± 0.49 days, higher in bilateral C.C.A. than unilateral (*p* = 0.003). Pain killer usage was required for an average of 2.2 ± 1.05, with significantly higher needs for bilateral C.C.A. (2.9 ± 1.13 days) than unilateral (1.73 ± 0.67) (*p* = 0.003). A significantly higher hospital stay was found for subjects with unilateral than bilateral C.C.A. (2.66 ± 1.39 vs. 9.8 ± 2.92; *p* < 0.001). In subgroups analysis between surgical revision patients, no statistical significance was reported for average hospital stay between stenosis >50% and <50% patients (*p* = 0.802) and postoperative pain (*p* = 0.075), while the surgical restenosis group demonstrated a higher delay in starting breast suction (*p* < 0.001). In the Kruskal–Wallis H test for healing subtypes, the patients reported no statistically significant differences in hospital stay length (*p* = 0.345) (Figure 2a) and postoperative pain (*p* = 0.256) (Figure 2b), while delayed breast suction was significant, being longer as the severity of the stenosis increases (Kruskal; *p* = 0.001) (Figure 2c).

Among the independent variables’ predictors of surgical success, the presence of Charge syndrome and rhinopharyngeal stenosis demonstrated the higher risks for surgical revision (OR: 4.00, 95% CI: 0.57–28.01, and OR: 2.75, 95% CI: 0.55–13.69, respectively) in the sample (Figure 3).

Conversely, an associated hypoplastic inferior turbinate and bilateral C.C.A. showed a lower risk for severe restenosis and surgical revision need (OR: 0.88, 95% CI: 0.22–3.52, and OR: 0.45, 95% CI: 0.10–2.08).

A high correlation expressed with the Jaccard similarity coefficient was found between the surgical revision and the presence of variables such as the presence of Charge syndrome (J = 0.71), rhinopharyngeal stenosis (J = 0.68), osseous C.C.A. (J = 0.63), and Male gender (J = 0.63) (Figure 4).

## 4. Discussion

Congenital choanal atresia is a rare craniofacial anomaly developing from a unilateral or bilateral membranous or bony obstruction malformation of the choana [3,5]. The main goal of the surgical approach is to prevent functional complications of the nose and upper jaw and reduce the risk of recurrence and the surgical revision need. Common recommendations are to obtain an adequate widening of the neochoana using mucosal flaps that adhere to the uncovered walls through fibrin glue [20,21,22,23,24].

Different preoperative parameters have been analyzed in the literature in order to predict the outcome of surgical treatment in C.C.A. Velegrakis et al., in a 28-year retrospective study on outcomes of endonasal surgery for C.C.A., associated the presence of purely bony atretic plaque and CHARGE syndrome significantly associated with disease recurrence (*p* < 0.001 and *p* = 0.049, respectively) while bilateral AC (*p* = 0.085) showed slight but not statistically significant correlations [24]. Our analysis among the independent predictive variables for surgical success reported an OR of 2.20 [0.48–10.07] and a high correlation in the Jaccard test (J = 0.63).

The stenting role in minimizing restenosis risk is still controversial, possibly leading to infection granulation, scar formation, and restenosis [25,26,27].

Several authors have asserted that the stents’ usage should be necessary for successful repair [28,29]. The stenting advantages are represented by the stabilization of the mucosal flaps and remodeling of the nasal wall.

Cumberworth et al. discussed how using nasal stents could cause various complications such as septal and intranasal synechia and a higher infection rate with resistant bacteria proliferation due to long-term antibiotic therapy [30]. Moreover, the stent could lead to granulation formation to the neochoana edge, needing medications and eventual revision surgery.

El-Ahl et al., treating 7 patients with bilateral C.C.A., reported adequate functional nasal breathing in all patients during a follow-up of 11 to 23 months. The authors associated stentless surgery with better visualization of the neochoana and an early recovery rate avoiding related complications. A systematic review investigated whether stenting could lead to better outcomes than stentless repair, identifying 48 research studies [31]. Although there is a lack of high-quality evidence, the authors concluded that stentless repair might experience fewer complications in bilateral C.C.A.

Our analysis showed that nasal stenting is associated with a higher risk of restenosis than the stentless technique, having been more frequent in cases of severe stenosis and surgical revision (OR: 1.36; 95% CI, 0.35–5.38) with a high similarity index (J = 0.55).

On the other hand, bilateral C.C.A. was also a protective factor for restenosis (OR: 1.36; CI 95%, 0.35–5.38), with a low similarity index (J = 0.47). However, it can be one of the signs of some malformation syndromes, the most common of which is CHARGE syndrome (coloboma, heart defects, atresia choanae, growth retardation, genitourinary abnormalities, and ear/deafness abnormalities) [20]. In 2008, a meta-analysis that grouped 20 studies and 238 cases of endonasal endoscopic approach for the treatment of choanal atresia stated restenosis was found more frequently in children with combined bilateral bone atresia multiple congenital disabilities, in particular Charge Association [28]. We have reported a higher risk of severe stenosis for patients with Charge syndrome than for non-syndromic subjects (OR: 4.00; CI 95% 0.57–28.01). Brihaye et al. in 2017 reported that in patients with hypoplasia of the lower turbinates, it was possible to use endoscopes with a larger diameter (4 mm) and a consequent better surgical field compared to normal inferior turbinate [3]. Our analysis reported a lower risk of patients with hypoplastic inferior turbinate, possibly due to better endoscopic access and a wider visualization of the surgical field (OR: 0.88; CI 95% 0.22–3.52; J = 0.45). In the literature, the preventive role of different variables such as immediate postoperative extubation, the early initiation of breast suction, and shorter hospital stay to promote quick restoration, increase upper airway flow, and reduce nasal congestion and fibrosis genesis is debated [26,27,28,29,30,31,32]. In the subgroups comparison according to surgical restenosis, we found a significant reduction in the delayed breast suction in <50% of stenosis cases compared to >50% (*p* < 0.001). Instead, the hospital stay difference was not significant in the subgroup comparison (*p* > 0.05) (Figure 2b). Also, in the Kruskal–Wallis test, only delayed breast suction demonstrated a statistically significant correlation with healing outcomes (*p* < 0.001). Instead, no significant correlation was demonstrated between healing grade and pain killer usage and hospital stay (*p* > 0.05 for both). The rarity of the disease has resulted in a small cohort of patients, although they have been enrolled for many years and in different hospital centers. The small sample reduced the statistical value of the risk factors analyzed. Furthermore, given that the technique used did not include two homogeneous groups treated with or without stents, the results should still be considered cautiously.

## 5. Conclusions

Current surgical approaches for C.C.A. should minimize the tendency for restenosis, shortening hospital stay, accelerating recovery and reducing postoperative pain. Several independent variables could influence surgical success after endoscopic repair of C.C.A., allowing the surgeon to choose the right surgical approach and appropriate postoperative follow-up. However, further high-quality evidence is needed to generate an effective predictive model.

## Figures and Tables

**Figure 1 ijerph-19-09084-f001:**
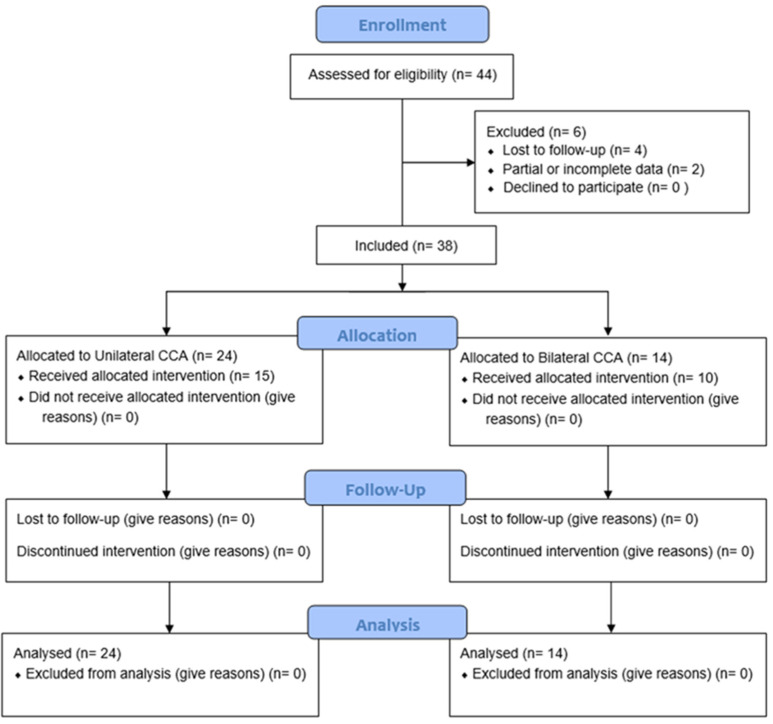
CONSORT 2010 Flow Diagram.

**Figure 2 ijerph-19-09084-f002:**
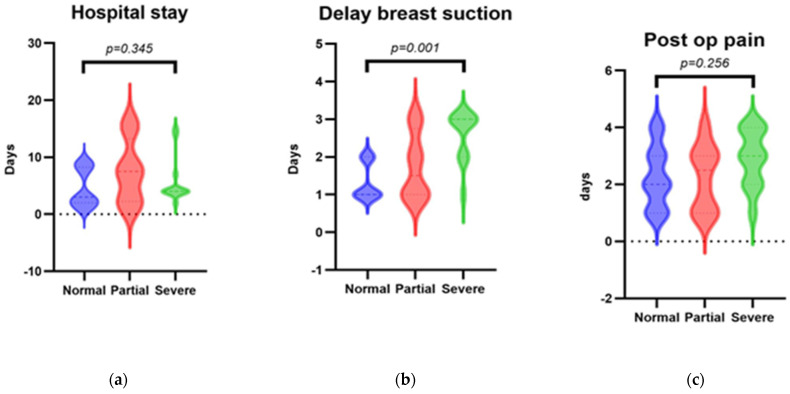
Post operative outcomes analysis divided for healing subtypes. In the Kruskal–Wallis test, a statistical significance was demonstrated only for delay in breast suction according to healing outcomes (*p* = 0.001).

**Figure 3 ijerph-19-09084-f003:**
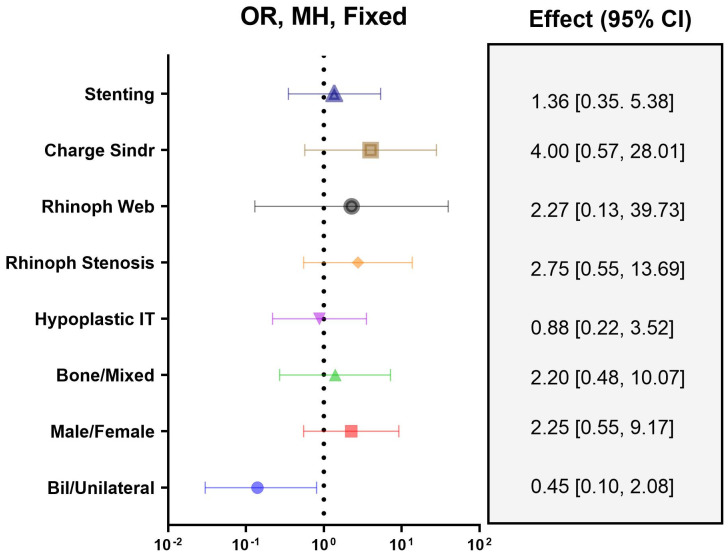
Forest–plot of independent variables for surgical success.

**Figure 4 ijerph-19-09084-f004:**
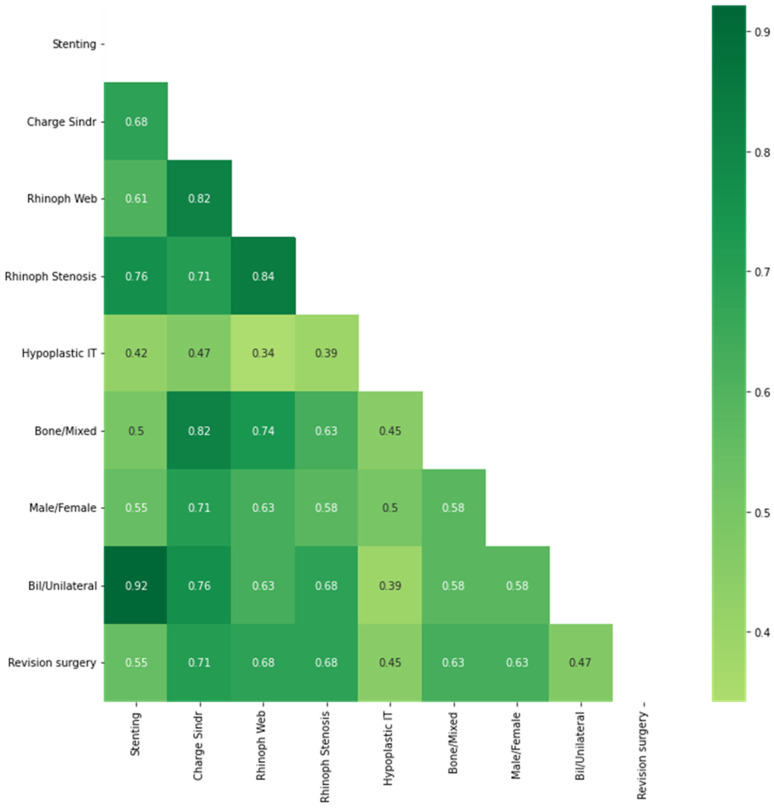
Heatmap of Jaccard similarity coefficients matrix.

**Table 1 ijerph-19-09084-t001:** The main demographic features are divided according to each choanal subgroup.

Features	Total	Bilateral	Unilateral	*p*-Value
Total number of cases	38 (100%)	14 (36.84%)	24 (63.16%)	
Gender	14 M (40%)/24 F (60%)	6 M (40%)/8 F (60%)	8 M (40%)/16 F (60%)	-
Type of atresia	8 B (12%)/30 BM (88%)	3 B (10%)/11 BM (90%)	5 B (13%)/19 BM (87%)	0.50
Age (days)		7.9 ± 1.3 days	105.5 ± 51.33 days	<0.001
Associated loco regional birth defects	33 (100%)	14 (40%)	19 (60%)	0.1
Hypoplastic concha inferior	23 (76%)	7 (20%)	16 (56%)	<0.001
Rhinopharyngeal stenosis	8 (20%)	5 (16%)	3 (4%)	0.01
Rhinopharyngeal web	2 (4%)	2 (4%)	-	-
Syndromic cases (Charge)	5 (12%)	5 (12%)	-	-
Revision surgery	12 (8%)	3 (4%)	9 (4%)	-
Delay for starting Breast-suction (days)	1.2 ± 0.49	1.55± 0.66	1	0.003
Duration of Pain (days)	2.2 ± 1.05	2.9 ± 1.13	1.73 ± 0.67	0.003
Hospital stays (days)	5.52 ± 4.09	9.8 ± 2.92	2.66 ± 1.39	<0.001
Healing postoperative				
Normal	18 (68%)	7 (24%)	11 (44%)	0.804
limited scar-partial restenosis (<50%)	8 (20%)	4 (12%)	4 (8%)	0.385
severe restenosis (>50%)	12 (8%)	3 (4%)	9 (4%)	0.303

## Data Availability

Not applicable.

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
