# Peer review of "Endoscopic Endonasal Repair of Congenital Choanal Atresia: Predictive Factors of Surgical Stability and Healing Outcomes"

_ijerph, 2022, doi:10.3390/ijerph19159084_

Round 1

Reviewer 1 Report

Choanal atresia is always an interesting topic and its treatment requires a meticulous approach. Therefore, I commend the authors for choosing this topic. However, I think the manuscript requires a lot of changes. I have listed some:

·        line 23 and 24: „All the patients were clinically and radiologically phenotyped and followed for at least 2 years.“

-        it is not very clear what kind of phenotyping the authors mean

·        line 33: „while Hypoplastic inferior turbinate and bilateral C.C.A. showed a lower“

-        why upper case for Hypoplastic

·        line 55: „there is no standard surgical protocol“

-        I think it should be mentioned that nowadays most CCA are operated endoscopically

·        line 69: A retrospective multicentric study was performed at our ear, nose, and throat (E.N.T.) Unit

-        a multicentric study carried out at a single center???

·        line 83 and 84: „The diagnosis was thus confirmed via clinical and endoscopic nasal assessment. Pre-operative high-resolution spiral C.T. scans were performed through the bone algorithm“

-        after failure to pass a catheter through the nasal caivity into the oropharynx the diagnosis of choanal atresia was made by endoscopic examination and confirmed by CT

·        line 104 and 105: microdebrider and drill

-        add size and manufacturer as for endoscope - line 95

·        line 132: while restenosis < 4 mm according to the endoscope diameter

-        the size of the koana is related to the weight of the child, that is, it is not always the same. A koana with a diameter of less than 4 mm can be of normal size

·        line 163 and 164: A total of 38 patients were included in the analysis, of which 24/38 (%) isolated unilateral C.C.A. while 14/38 (%) had bilateral C.C.A.

-        percentages without number

·        line 166: Hypoplastic inferior turbinate

-        why upper case?

-        how do you define hypoplastic inferior turbinate in the newborn

·        line 219 and 220: prevalence of approximately 1:5000-8:000 per live birth

-        you said that in the first sentence of Introduction

I think that before submission, the manuscript should be reviewed once again by the author's senior colleague experienced in this field.

Author Response

Revisor 1

Comment:   line 23 and 24: „All the patients were clinically and radiologically phenotyped and followed for at least 2 years.“

-        it is not very clear what kind of phenotyping the authors mean

  • Response: dear revisor, we corrected the term in assessed as the management included both clinical and radiological evaluation.

Comment:       line 33: „while Hypoplastic inferior turbinate and bilateral C.C.A. showed a lower“

-        why upper case for Hypoplastic

Response: Hypoplastic turbinate, as discussed later, has been identified as a positive surgical outcome due to the more space characterized during a surgical procedure.

We added the following sentence to better clarify: by an higher endoscopic surgical space and creating a single larger opening’’

  • Comment: line 55: „there is no standard surgical protocol“

-        I think it should be mentioned that nowadays most CCA are operated endoscopically

Response: as suggested, we changed the sentence to highligh the role of miniinvasive tecniques commonly performed nowdays.

     Comment:          line 69: A retrospective multicentric study was performed at our ear, nose, and throat (E.N.T.) Unit

-        a multicentric study carried out at a single center???

Response: thanks for the suggestion, we’ve corrected the mistake in multicentric study.

  • Comment: line 83 and 84: „The diagnosis was thus confirmed via clinical and endoscopic nasal assessment. Pre-operative high-resolution spiral C.T. scans were performed through the bone algorithm“

-        after failure to pass a catheter through the nasal caivity into the oropharynx the diagnosis of choanal atresia was made by endoscopic examination and confirmed by CT

Response: thanks for the suggestion, as indicated we’ve added the concept to the sentence

  • Comment: line 104 and 105: microdebrider and drill

-        add size and manufacturer as for endoscope - line 95

Response: dear editor, as indicated we’ve added both the informations required.

  • Comment: line 132: while restenosis < 4 mm according to the endoscope diameter

-        the size of the koana is related to the weight of the child, that is, it is not always the same. A koana with a diameter of less than 4 mm can be of normal size

Response: thanks for the suggestion, we’ve removed the sentence and only maintained the different restriction divided in subclasses.

  • Comment: line 163 and 164: A total of 38 patients were included in the analysis, of which 24/38 (%) isolated unilateral C.C.A. while 14/38 (%) had bilateral C.C.A.

Response: thanks for the suggestion, we’ve corrected the mistakes

  • Comment: line 166: Hypoplastic inferior turbinate

-        why upper case?

-        how do you define hypoplastic inferior turbinate in the newborn

Response: The most frequently associated loco-regional disorders in the literature hypoplasia of the inferior turbinate is present in up to 70% of cases and can occur monolaterally or biltaterally. It is defined as reduced pneumatization of the inferior concha that is evidenced endoscopically and by CT in volumetric reduction of the inferior turbinate by at least 50% compared with the contralateral if normal size. Our data were concordant with the data in literature (doi:10.1016/j.ijporl.2017.04.022), we’ve added the reference also in the methods were we explained all the features evaluated during the management.

  • Comment: line 219 and 220: prevalence of approximately 1:5000-8:000 per live birth

-        you said that in the first sentence of Introduction

Response: thanks, repetitive sentence was removed.

Reviewer 2 Report

The authors examined 64 cases of the perceived postoperative functional and subjective outcomes in a series of congenital choanal atresia patients, comparing the results obtained in the unilateral forms with the bilateral ones. The topics are interesting and meet the academic demands. However, there are several major points which should be amended before considering publication.

1.    Difference in Surgical procedure between bilateral and unilateral:

“The enlargement of the bone frame was continued with the drill up to the sphenoid rostrum at the top and the pharyngeal wall on the outside.”

It is obvious that bilateral CAA cases require immediate surgery for life saving. Therefore, both the methods and bony structure should be different. The authors should describe more in clear fashion.

2.    The mean length of postoperative stenting.

The authors should clarify the criteria how long to place the stenting tubes. How to set up the timing to remove. The composition of the stent should be described more in detail.  

3.    I understand that the authors evaluated healing postoperative courses by using CT images. The processes of image analysis by CT scanning should also be presented.

4. The authors should describe how to manage Rhinopharyngeal stenosis 8 (20%) 5 (16%) 3 (4%) and Rhinopharyngeal web in Table 1.

5.    LONG-TERM OUTCOMES AND SURGICAL STABILITY

The title depicts long-term outcomes. However, there are no data as for factors related to long-term conditions.

6.    A recent systematic review investigated whether stenting could lead to better outcomes than stentless repair, identifying 48 research studies [31].

Ref. 31 is not a resent review.

Author Response

Revisor 2

Comment: 1. Difference in Surgical procedure between bilateral and unilateral:

“The enlargement of the bone frame was continued with the drill up to the sphenoid rostrum at the top and the pharyngeal wall on the outside.”

It is obvious that bilateral CAA cases require immediate surgery for life saving. Therefore, both the methods and bony structure should be different. The authors should describe more in clear fashion.

Response:  thanks for the suggestions, we’ve added the following sentences to describe clearly the tecniques ‘’ Since bilateral forms lead to neonatal respiratory distress, management has included the use of an oropharingeal (Guedel type) or orotracheal intubation. In all cases of single- or bilateral C.C.A.

The posterior nasopharynx was protected during the maneuver with cotton inserted through the contralateral choana or oropharynx of the patient and to avoid possible subperiosteal detachment of the posterior cul-de-sac during perforation of the atresal wall

Enlargement of the bony structure was continued with the drill up to the roof of the nasopharynx and the sphenoidal rostrum at the top while laterally bringing the pterygopalatine block at the level of the lateral nasopharyngeal wall; however, careful attention was performed to the area of the sphenopalatine foramen to avoid injuring the sphenopalatine artery’’.

Comment: 2.    The mean length of postoperative stenting.

The authors should clarify the criteria how long to place the stenting tubes. How to set up the timing to remove. The composition of the stent should be described more in detail. 

Response: thanks for the suggestions, we distinguished between unilateral and bilateral stenting, materials, and motivated the criteria to remove the stent and determine the correct time.

Comment: 3. I understand that the authors evaluated healing postoperative courses by using CT images. The processes of image analysis by CT scanning should also be presented.

Response: thanks for the suggestions, we’ve added the algorithm performed ‘’A C.T. section parallel to the posterior hard palate at the level of the pterygoid plates was obtained in all children. The sections were 0.5 to 1 mm thick. The width of the vomer at the posterior end and the width of the airspace of the choanae or membranous atresia were measured based on the measurement of the dimension from the lateral wall of the nasal cavity to the edge of the vomer of the posterior end.’’

Comment: 4. The authors should describe how to manage Rhinopharyngeal stenosis 8 (20%) 5 (16%) 3 (4%) and Rhinopharyngeal web in Table 1.

Response: thanks for the suggestions, we’ve added the description to improve the quality ‘’ In the presence of rhinopharyngeal stenosis or web under endoscopic visualization, a flap knife was used to cut the scarred mucosa on the posterior septum, on the posterior end of the inferior or middle turbinate, then excising with shaver and Blakesley's  forceps.’’

Comment: 5.    LONG-TERM OUTCOMES AND SURGICAL STABILITY

The title depicts long-term outcomes. However, there are no data as for factors related to long-term conditions.

Response: dear revisor, thanks for the suggestion, we’ve modified the title as indicated in ‘’ ENDOSCOPIC ENDONASAL REPAIR OF CONGENITAL CHOANAL ATRESIA: PREDICTIVE FACTORS OF SURGICAL STABILITY AND HEALING OUTCOMES’’

Comment: 6.    A recent systematic review investigated whether stenting could lead to better outcomes than stentless repair, identifying 48 research studies [31].

Ref. 31 is not a resent review.

Response: thanks for the suggestions, we’ve removed the term.

Reviewer 3 Report

The authors propose an interesting paper on surgical management of choanal atresia. The study is multicentric,fundamental for rare diseses,  well designed and written. In the discussion there's a complete but essential review of litterature on the most intriguing points. 

I would like to make only a few comments:

line 90-91: I suggest not to repeat twice  exclusion criteria in few lines (lines 85-86)

Methods: Has pre-operative or post-operative prophylaxis been performed?  Particularly, the presence/absence of stenting has influenced the antibiotic choice? (line 239-240)

Discussion: I'm aware that the results are conditioned by the rarity of the disease, as correctly stated by the authors themselves, but do the authors thinks that it could be possible to express a stronger position on the use  vs non use of stenting in bilateral CCA?

Author Response

Comment: line 90-91: I suggest not to repeat twice exclusion criteria in few lines (lines 85-86)

Response: thanks for the suggestion, we’ve removed the repetitive informations.

Comment: Methods: Has pre-operative or post-operative prophylaxis been performed?  Particularly, the presence/absence of stenting has influenced the antibiotic choice? (line 239-240)

Response: thanks for the suggestions, we’ve not distiunguished among patients with stent or not in post operative antibiotic treatment. ‘’All patients received postoperative antibiotic prophylaxis for 10 days after surgery, regardless of stent use’’.

Comment: Discussion: I'm aware that the results are conditioned by the rarity of the disease, as correctly stated by the authors themselves, but do the authors thinks that it could be possible to express a stronger position on the use vs non use of stenting in bilateral CCA?

Response: thanks for the suggestions, we’ve modified the following sentence to debate the role of nasal stenting increasing revision risk and restenosis. ‘’ Our analysis showed that nasal stenting is associated with a higher risk of restenosis than the stentless technique, having been more frequent in cases of severe stenosis and surgical revision (OR:1.36; 95% CI, 0.35-5.38) with a high similarity index (J=0.55).’’

Round 2

Reviewer 1 Report

             line 35: „theHypoplastic inferior turbinate and bilateral“ - why upper case for Hypoplast

Author Response

Comment line 35: the Hypoplastic inferior turbinate and bilateral “- why upper case for Hypoplast’’

Response: thanks for the suggestions, we’ve corrected the paper

Reviewer 2 Report

Comment: 3. The processes of image analysis by CT scanning should also be presented.

Response: thanks for the suggestions, we’ve added the algorithm performed ‘’A C.T. section parallel to the posterior hard palate at the level of the pterygoid plates was obtained in all children. The sections were 0.5 to 1 mm thick. The width of the vomer at the posterior end and the width of the airspace of the choanae or membranous atresia were measured based on the measurement of the dimension from the lateral wall of the nasal cavity to the edge of the vomer of the posterior end.’’

I confirmed that the authors described the algorithm. However, there are no data in the result section. A brief summary related to differences between bilateral and unilateral CAAs in the healing process is necessary.  

Author Response

Reviewer 2

Comment: I confirmed that the authors described the algorithm. However, there are no data in the result section. A brief summary related to differences between bilateral and unilateral CAAs in the healing process is necessary. 

Response: thanks for the suggestions, we’ included the following paragraph. ‘’Postoperative healing was overall normal at follow-up in 18/38 (68%) patients, including 7/38 (24%) bilateral and 11/37 (44%) unilateral C.C.A. cases (p=0.804). A nonstatistical difference was also recorded for limited scar formation-partial restenosis (<50%), found in 8/38 (20%) subjects, including 4 (12%) bilateral and 4 (8%) unilateral (p = 0.385). In addition, severe restenosis (>50%) was reported in 12 (8%) cases, including 3 (4%) bilateral and 9 (4%) unilateral (p=0.303), finally requiring surgical revision.’’
